# The Mental Health of Young Return Migrants with Ancestral Roots in Their Destination Country: A Cross-Sectional Study Focusing on the Ethnic Identities of Japanese–Brazilian High School Students Living in Japan

**DOI:** 10.3390/jpm12111858

**Published:** 2022-11-07

**Authors:** Eriko Fukui, Takashi Uchino, Masunari Onozaka, Takashi Kawashimo, Momoko Iwai, Youji Takubo, Akiko Maruyama, Sachio Miura, Ryo Sekizaki, Masafumi Mizuno, Naoyuki Katagiri, Naohisa Tsujino, Takahiro Nemoto

**Affiliations:** 1Department of Neuropsychiatry, Toho University Graduate School of Medicine, 5-21-16 Omori-nishi, Ota-ku, Tokyo 143-8540, Japan; 2Department of Neuropsychiatry, Toho University Faculty of Medicine, 6-11-1 Omori-nishi, Ota-ku, Tokyo 143-8541, Japan; 3Faculty of Nursing, Shoin University, 9-1 Morinosatowakamiya, Atsugi-shi 243-0124, Japan; 4School of Tropical Medicine and Global Health, Nagasaki University, 1-14 Bunkyo-cho, Nagasaki 852-8131, Japan; 5Kiryu Daiichi High School, 1-5 Kosone-cho, Kiryu-shi 376-0043, Japan; 6Tokyo Metropolitan Matsuzawa Hospital, 2-1-1 Kamikitazawa, Setagaya-ku, Tokyo 156-0057, Japan

**Keywords:** ethnic identity, mental health, migrant, transcultural psychiatry, youth

## Abstract

Background: The number of young Japanese Brazilians, who are return migrants with Japanese ancestral roots, is increasing rapidly in Japan. However, the characteristics of their mental health and the relation between mental health and a complex ethnic identity remains unclear. Methods: This cross-sectional study compared 25 Japanese–Brazilian high school students with 62 Japanese high school students living in the same area. Research using self-report questionnaires on mental health, help-seeking behavior tendencies, and ethnic identity was conducted. The Japanese–Brazilian group was also divided into high and low ethnic identity groups, and their mental health conditions were compared. Results: The Japanese–Brazilian group had significantly poorer mental health conditions and lower ethnic identities than the Japanese group and were less likely to seek help from family members and close relatives. Among the Japanese Brazilians, those with low ethnic identity had significantly poorer mental health than those with high ethnic identity. Conclusions: Young Japanese Brazilians may face conflicts of ethnic identity that can disturb their mental health. To build an inclusive society, the establishment of community services to support mental health and to help return migrants develop their ethnic identity is essential.

## 1. Introduction

The number of international migrants, estimated at 272 million worldwide, has been increasing, and the globalization of culture, economy, and politics has been accelerated. The United States and European countries have long accepted international migrants and refugees; in recent years, international migration has been increasing in Asia as well. Japan has the second largest number of foreign residents in Asia, with 2.76 million (2.2% of the total population) as of 2022 [1]. Furthermore, global migration has become complex and diverse, crossing not only national borders but also racial, ethnic, cultural, and generational boundaries. For example, some migrants have returned to their home countries (known as “return migration”). Among return migrants, some return migrants have ancestors who migrated internationally and have returned to their ancestors’ home country, i.e., they are “return migrants with ancestral roots in the destination country.” Migrants with ancestral roots in Japan are called “Nikkei” in Japanese. There are approximately 4 million Nikkei worldwide, with Japanese Brazilians comprising the largest number of Nikkei individuals. Many Japanese emigrated to Brazil as migrant workers between 1908 and 1993, and their descendants, who are second- and third-generation Japanese Brazilians, have returned to Japan to improve their careers and living standards. The number of Japanese Brazilians who have returned to Japan has dramatically increased since 1990, when a change in Japanese law eased restrictions on migrants’ work, and as of 2018, Japanese Brazilians comprise the second largest group of migrants in Japan, following migrants from Asia [2]. It is not only Japanese–Brazilian men who have returned to Japan for work, but also their entire families, resulting in a rapid increase in the number of young Japanese Brazilians. In fact, the largest age group of Japanese–Brazilian residents in Japan is the young generation.

Migration away from one’s home country can cause various stresses that can have a significant impact on an individual’s mental health and well-being [3,4]. Migration has been found to be a risk factor for the development of various mental illnesses, including stress-related disorders and psychotic disorders [5,6]. Migrants with mental illness are likely to experience suicidal ideation, and high suicide rates have been reported [7,8]. Furthermore, migration can cause trans-generational difficulties in their offspring, and migrants are reported to be more susceptible to stress due to various reasons [9,10]. However, around the world, including in Europe, North America, Oceania, and Asia, migrants have less access to appropriate mental health services than non-migrants, even though they are often in greater need of such services [11,12]. This tendency is particularly pronounced among young migrants in Japan [13]. The onset of mental illnesses during youth is relatively common, and youths are the most important age group for targeted early psychiatric intervention to prevent the onset of mental illness and to improve their prognosis [14,15,16,17]. Very little research has been conducted on community-dwelling young migrants that quantitatively assesses their mental health characteristics and help-seeking behaviors, despite the urgent need for such research because of the growing number of migrants.

Mental health problems among migrants can result from a complex interplay of biological, cultural, and social factors. For example, difficulties in adapting to different cultures, language barriers, inadequate access to economic or social resources, and discrimination and prejudice against migrants have been reported [18,19]. Changes in ethnic identity are recognized as an important within-individual factor related to the mental health of migrants [20]. Ethnic identity is defined as “an attachment to race and culture in the group to which one belongs.” Migrants often experience ethnic identity conflicts when they leave the culture in which they grew up and are exposed to a different culture [21]. Such ethnic identity conflicts can have negative impacts on mental health [22,23]. Conversely, a robust ethnic identity may act protectively against stress and depression [24].

As mentioned above, Japanese Brazilians are a major group of return migrants living in Japan who have ancestral roots in Japan. Considering the above, the aim of the present study was to investigate the characteristics of mental health among young Japanese Brazilians living in Japan, focusing on their ethnic identity.

## 2. Materials and Methods

### 2.1. Study Design and Participants

This study was a cross-sectional, questionnaire-based study of Japanese–Brazilian high school students and Japanese high school students living in the same area. Surveys were conducted at a rural high school for Japanese Brazilians in December 2021 and March 2022 and at a public high school in the same geographical area in November 2021. Of the 26 Japanese–Brazilian high school students and the 70 Japanese high school students who were candidates for this study, 25 Japanese–Brazilian high school students and 62 Japanese high school students participated.

This survey was conducted anonymously. The Japanese high school students and their parents were provided with manuals and questionnaires written in Japanese, while the Japanese–Brazilian students and their parents were given versions written in Portuguese. Portuguese translation specialists translated all the questionnaires and scales used in this study from Japanese to Portuguese. The translated questionnaires were confirmed to have no semantic differences among several Portuguese–Japanese bilingual speakers who were not involved in the translation. Since the subjects were regarded as comprising a vulnerable population, we emphasized prior to participation that the subjects and their parents could refuse to participate in the study with no adverse consequences. In addition, before conducting the questionnaires, the informed consent of the subjects was confirmed in writing. The study protocol was approved by the Ethics Committee of the Faculty of Medicine, Toho University (approval number: A21004). This study was performed in accordance with the latest version of the Declaration of Helsinki.

### 2.2. Measures

Various demographic measures, including age, sex, nationality, sense of belonging, years of residence in Japan, and economic conditions, were investigated. To assess the economic conditions, we asked the question, “Does your family own a car?” To examine the mental health conditions of the subjects and their levels of psychological distress, the WHO-Five Well-being Index (WHO-5J) and the Kessler 6 scale (K6) were used [25,26]. The WHO-5J consists of 5 items rated using a six-point Likert scale (total range of 0 to 25, with a lower score indicating a poorer mental health condition). The K6 consists of 6 items rated using a five-point Likert scale (total range of 0 to 24, with a higher score indicating greater psychological distress). Both scales examine the subject’s condition during the last two weeks. The cutoff scores of the WHO-5J and K6 are 12/13 and 4/5, respectively. The General Help-Seeking Questionnaire (GHSQ) was used to assess the tendency to seek help for psychological problems [27]. It consists of 10 items rated using a seven-point Likert scale that measures the likelihood of seeking help from different types of help providers when a mental health problem arises. The GHSQ consists of two sub-scales: informal resources, such as parents and friends; and formal resources, such as medical doctors and counselors (each sub-scale ranges from 0 to 30, with a lower score indicating a lower tendency of seeking help). The Multigroup Ethnic Identity Measure (MEIM) was used to assess cross-cultural adaptation and belonging [28]. Although the original version consists of 14 items rated using a four-point Likert scale, in this survey, a simplified two-point Likert scale version was used (total range of 0 to 14, with a lower score indicating a lower ethnic identity).

### 2.3. Data Analysis

First, to compare the Japanese–Brazilian and Japanese high school students, an independent *t*-test was used for continuous variables and a chi-square test was used for categorical variables. Next, the group of Japanese–Brazilian high school students was divided into a high ethnic identity group and a low ethnic identity group using the median of the MEIM total score as a threshold, and the WHO-5J and K6 scores of the two subgroups were compared using independent *t*-tests. Statistical significance was set at 5% (*p* < 0.05). The statistical analyses were conducted using SPSS, version 26.0 (IBM, Armonk, NY, USA).

## 3. Results

The demographics of the participants are shown in Table 1. In the Japanese–Brazilian group, 36.0% (9/25) were male, the mean age was 16.2 years (SD = 1.2), and 92.0% (23/25) were of Brazilian nationality. In the Japanese group, 56.6% (35/62) were male, the mean age was 16.6 years (SD = 1.1), and 100% (62/62) were of Japanese nationality. In both groups, all the participants’ families owned a car. No significant differences in sex ratio, age, or economic status were seen between the Japanese–Brazilian and Japanese groups. No significant differences in sex ratio, age, or economic status were seen between the Japanese–Brazilian and Japanese groups. Regarding the sense of belonging, 100% (62/62) of the Japanese group answered, “*I am Japanese*”, while in the Japanese–Brazilian group, 12.0% (3/25) answered “*I am Japanese*” and 88.0% (22/25) answered “*I am Brazilian*”.

Next, the WHO-5J, K6, GHSQ, and MEIM scores were compared between the Japanese–Brazilian and the Japanese groups (Figure 1). The mean WHO-5J score in the Japanese–Brazilian group was 11.20 (SD = 4.72), which was significantly lower than that in the Japanese group (15.85, SD = 4.08, *p* < 0.001). The mean K6 score in the Japanese–Brazilian group was 8.48 (SD = 5.13), which was significantly higher than that in the Japanese group (4.5, SD = 3.87, *p* = 0.001). The mean GHSQ total score in the Japanese–Brazilian group was 15.80 (SD = 8.99), which was significantly lower than that in the Japanese group (21.40, SD = 7.58, *p* = 0.004). The mean GHSQ sub-scale for informal resources in the Japanese–Brazilian group was 8.92 (SD = 4.86), which was significantly lower than that in the Japanese group (13.58, SD = 5.41, *p* < 0.001). However, no significant difference in the mean GHSQ sub-scale for formal resources was seen between the Japanese–Brazilian group (4.60, SD = 5.40) and the Japanese group (4.13, SD = 3.96, *p* = 0.654). The mean MEIM score in the Japanese–Brazilian group was 6.20 (SD = 3.38), which was significantly lower than that in the Japanese group (7.65, SD = 2.72, *p* = 0.040).

Finally, the Japanese–Brazilian group was divided according to the median of the MEIM total score. The median MEIM score in the Japanese–Brazilian group was 6 points. Twelve people with a total score of 7 or higher were classified as belonging to a high ethnicity group, while 13 people with a total score of 6 or lower were classified as belonging to a low ethnicity group (Table 2). The mean WHO-5J score in the low ethnicity group was 9.31 (SD = 5.11), which was significantly lower than that in the high ethnicity group (13.25, SD = 3.36, *p* = 0.034). No significant difference in the mean K6 score was seen between the Japanese–Brazilian group (10.38, SD = 5.38) and the Japanese group (6.42, SD = 4.12, *p* = 0.051).

## 4. Discussion

In the present study, we examined the characteristics of mental health and help-seeking behavior tendencies of Japanese–Brazilian high school students by comparing them with Japanese high school students living in the same rural area. In addition, we examined the mental health conditions of Japanese Brazilians from the perspective of ethnic identity.

First, the Japanese–Brazilian high school students had significantly poorer mental health conditions than the Japanese high school students. Migrants from other home countries reportedly experience various stresses [3,4,5,6], and similar results were found in the present study examining return migrants with ancestral roots in Japan. Return migrants often return to their ancestral country of their own volition, but many Japanese–Brazilian high school students return regardless of their volition because of family circumstances, such as employment, economic reasons, or parental divorce [29]. Therefore, they are considered to represent a vulnerable population. In Japan, young migrants are less likely to access mental health care services than expected based on population demographics alone [13]. Thus, the present results suggest that young return migrants may have a lower likelihood of receiving adequate care and support at an early stage, despite being at an increased risk of experiencing mental health problems.

Regarding the tendency to seek help, Japanese Brazilians were less likely to seek help from informal resources (family members and friends) in the present study, although a previous study reported the opposite results [30]. One reason for this finding is presumed to be related to the fact that all the parents and guardians of the present subjects were employed and had limited opportunities to communicate with their children. This type of living situation makes it more difficult for youths to ask for help [21]. In addition, Japanese Brazilians often dwell in small, closed communities, potentially making it difficult for youths to seek help from friends or family who might be prone to spreading private information. This situation seems to reflect the reality of Japanese–Brazilian families dwelling in Japan. On the other hand, no significant difference in the tendency to seek help from formal resources (medical doctors and counsellors) was seen between the Japanese–Brazilian and the Japanese groups. This finding suggests that the development of community-based, integrated mental health care systems for migrants is important for providing easy access to help as well as a place where young people can feel comfortable seeking help [14,31,32]. Access to a social support network is an important factor in promoting good mental health. Furthermore, since many Japanese Brazilians living in Japan cannot speak Japanese, the development of appropriate services, including medical interpreters with experience in the fields of mental health and psychiatry, is needed.

In terms of ethnic identity, the identity level of Japanese–Brazilian high school students was significantly lower than that of Japanese high school students. This finding suggests that Japanese–Brazilian high school students have experience conflicts regarding their ethnic identity. Leaving the culture and environment where one was born and raised and coming into a society with a different culture can easily cause conflicts of ethnic identity. Furthermore, migration to an ancestral home country may further complicate ethnic identity [33]. In the present results, the subjects with lower ethnic identities had poorer mental health scores than those with higher ethnic identities. Japanese Brazilians living in Japan could be a unique example of return migrants in that they have a weak sense of belonging to both Japan and Brazil. When people adapt to different cultures, they undergo acculturation [34]. However, return migrants with roots in the home country of their ancestors feel an affinity for the two different cultures. In the process of cross-cultural adaptation, friction and conflicts could arise between the culture of one’s ancestral roots and one’s own culture, which may hinder identity formation. They could have a more complex background than other international migrants [35], and further research is needed. Support to help such individuals develop a healthy ethnic identity may be effective for preventing psychological problems. Since ego identity, including ethnic identity, develops during adolescence, approaches to encourage such development are desirable.

An important limitation of this study is that the participants were only 25 Japanese–Brazilian high school students, all of whom attend one school in one city. Although the area where this school is located has one of the largest Japanese–Brazilian populations in Japan, we should be cautious about generalizing the results of this study as characteristics of the entire return migrants in Japan. Additionally, the causal relationship between mental health and ethnic identity cannot be proven, given this study’s cross-sectional design. Further research examining a larger sample is expected in the future.

In addition, careful discussion is needed regarding whether each scale is sufficiently valid for the present sample. Since we have not used other scales in this study, it is difficult to test the validity of the scales further, although all the scales used in this study are used worldwide and have already been validated well [27,36,37,38]. Considering this important limitation, we would like to conduct future studies including an investigation of the scales’ validity for Japanese Brazilians.

## 5. Conclusions

The Japanese government is actively promoting policies to accept migrants to cover labor shortages caused by the aging society in Japan. The number of migrants, especially young adults, is expected to increase further. This study revealed the present mental health conditions of Japanese Brazilians living in Japan, and the present findings are expected to be a first step toward efforts to establish a convivial and inclusive society.

## Figures and Tables

**Figure 1 jpm-12-01858-f001:**
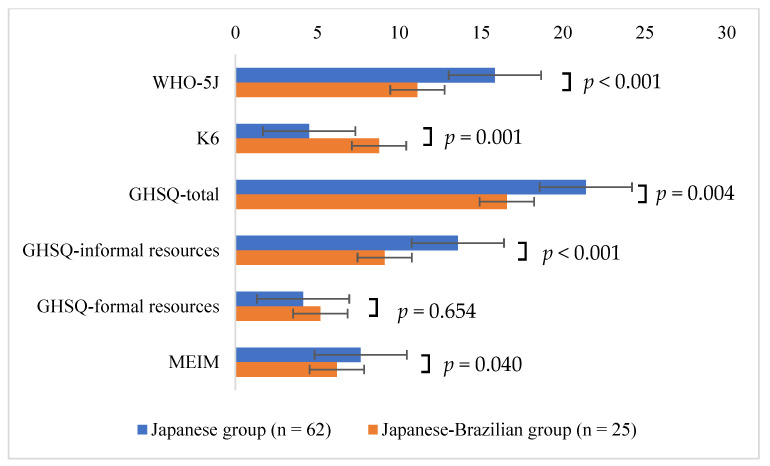
Comparisons of WHO-5J, K6, and GHSQ results between Japanese and Japanese–Brazilian high school students. WHO-5J: the WHO-Five Well-being Index, K6: the Kessler 6 scale, GHSQ: the General Help-Seeking Questionnaire, MEIM: the Multigroup Ethnic Identity Measure.

**Table 1 jpm-12-01858-t001:** Demographics of the participants.

		Japanese (n = 62)	Japanese Brazilian (n = 25)	*p*-Value
Age	Mean (SD)	16.6 (1.1)	16.2 (1.2)	0.108
Gender	M/F; n (%)	35 (56.5)/27 (43.5)	9 (36.0)/16 (64.0)	0.123
Owning a car	Yes/No (%)	62 (100)/0 (0)	25 (100)/0 (0)	-
Nationality				
Japan	n (%)	62 (100.0)	-	
Brazil	n (%)	-	23 (92.0)	
Japan and Brazil	n (%)	-	2 (8.0)	
Sense of belonging	Japanese/ Brazilian; n (%)	62 (100)/0	3 (12.0)/22 (88.0)	<0.001

SD: standard deviation; M: males; F: Females.

**Table 2 jpm-12-01858-t002:** Comparisons of WHO-5J and K6 scores between high and low ethnicity groups (n = 25).

	High Ethnicity Group n = 12	Low Ethnicity Group n = 13		
	Mean (SD)	Mean (SD)	*p*-Value	Cohen’s d *
WHO-5J total score	13.25 (3.36)	9.31 (5.11)	0.034	0.91
K6 total score	6.42 (4.12)	10.38 (5.38)	0.051	0.83

* Cohen’s d, d = (M1−M2)/√ ((SD1^2^ + SD2^2^)/2). M1 = the mean score of the High ethnicity group, M2 = the mean score of the Low ethnicity group. SD1 = the standard deviation of High ethnicity group, SD2 = the standard deviation of the Low ethnicity group.

## Data Availability

Data supporting the findings of this study are available upon reasonable request to the corresponding author. The data will not be made publicly available because of privacy and ethical restrictions.

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
