# Peer review of "The Mental Health of Young Return Migrants with Ancestral Roots in Their Destination Country: A Cross-Sectional Study Focusing on the Ethnic Identities of Japanese–Brazilian High School Students Living in Japan"

_jpm, 2022, doi:10.3390/jpm12111858_

Round 1

Reviewer 1 Report

This article focuses on migration in the context of complex and diverse globalization. Using a cross-sectional study, it examines the mental health of Japanese Brazilian high school students and the relationship between mental health and ethnic identity. The topic of return migration is innovative and interesting.

But before publishing, there are still some problems:

1. The study was conducted in only one rural secondary school with a small sample size; thus this study has a large sampling problem. The findings of the study have large limitations.

2. migrants' health issues are influenced by social determinants such as economic status, residence, and social networks etc. The authors have mentioned they had collected economic variables, but the article lacks a description of the content.

3. The questionnaires/scales used in the study were in different languages. When creating questionnaires/scales in different languages, how did the study avoid the effect of different languages on the meaning of the questionnaires/scales? 

Author Response

Please find attached the file.

Reviewer 2 Report

I find the selected topic very necessary and interesting. Congratulations.  

The summary is well structured, clear and concise.

The references in the introduction seem to me to be somewhat scarce for an article of this scientific rigour.  I recommend including more studies from 2021 and 2022.

When you say "It is believed that they have more unique and complex antecedents..." in what sense? It is not entirely clear at this point.  

In terms of method, is the questionnaire you used, is it self-developed, is it validated? is it self-developed, is it validated? In my opinion, this is the weak point of the article that needs to be article that needs to be significantly improved. Even if scales from other authors have been used, it is scales of other authors have been used, they need to be validated in your sample.

Author Response

Please find attached the file.

Round 2

Reviewer 1 Report

This article focuses on migration in the context of complex and diverse globalization. Using a cross-sectional study, it examines the mental health of Japanese Brazilian high school students and the relationship between mental health and ethnic identity. The topic of return migration is innovative and interesting.

The authors have given reasonable answers to previous questions. So I recommend publishing the article.

Reviewer 2 Report

I consider that the article has improved significantly. The authors have correctly responded to the suggestions